# Transcriptomic Analysis Reveals Patterns of Expression of Stage-Specific Genes in Early *Apis cerana* Embryos

**DOI:** 10.3390/genes16020187

**Published:** 2025-02-03

**Authors:** Runlang Su, Yuhui Chen, Rui Zhu, Guiling Ding, Kun Dong, Mao Feng, Jiaxing Huang

**Affiliations:** 1College of Animal Science and Technology, Yunnan Agricultural University, Kunming 650201, China; surunlang@gmail.com (R.S.);; 2State Key Laboratory of Resource Insects, Chinese Academy of Agricultural Sciences, Beijing 100093, China; chenyuhui0902@163.com (Y.C.); 82101225470@caas.cn (R.Z.);; 3Key Laboratory for Insect-Pollinator Biology of the Ministry of Agriculture and Rural Affairs, Institute of Apicultural Research, Chinese Academy of Agricultural Sciences, Beijing 100093, China

**Keywords:** honeybees, *A. cerana*, worker, drone, embryonic transcriptomes

## Abstract

Background/Objectives: *Apis cerana* development is described as comprising four stages: embryo, larva, pupa, and adult. There are significant differences between workers and drones in terms of physiological functions and social roles, and the formation of the organ primordia occurs during the embryonic stage. Therefore, the objective of this study is to investigate the differential expression of and alternative splicing of genes in worker and drone embryos and to explain their unique developmental patterns. Methods: Long-read sequencing (PacBio Iso-Seq) and short-read sequencing (Illumina RNA-Seq) were used to investigate worker and drone embryo gene expression differences in *A. cerana* across five developmental points (12, 24, 36, 48, and 60 h). Results: The study identified 59,254 common isoforms, with 5744 and 5106 isoforms specific to worker and drone embryos, respectively. Additionally, a new transcript of the *csd* gene was identified. The number of differentially expressed genes (3391) and differential splicing events (470 genes) peaked at the 24-h embryonic stage. Differential splicing events of *csd*, *dsx*, and *Y-y* were observed in the worker and drone embryos. Conclusions: The gene expression results indicated that the 24-h embryonic point is a critical period for the expression of genes related to developmental and behavioral differences between workers and drones. The findings provide a theoretical basis for future research on the developmental differences between workers and drones.

## 1. Introduction

The life cycle of honeybees’ development includes four stages: egg, larva, pupa, and adult [1,2]. The embryo is the first and most critical stage, in which the basic organs of the honeybees are formed. Gene expression during this stage significantly influences the morphology, appearance, and behavior of adult honeybees, which is tightly controlled by environmental and intracellular signals [3]. The fertilized diploid embryos of honeybees generally develop into queens and workers, while unfertilized embryos develop into haploid males [4]. Although there are no phenotypic differences in the *A. cerana* Fabricius, 1793, embryos and throughout the first four larval stages [5,6], the developmental duration differs markedly for queens (16 d), workers (20 d), and drones (23 d) under normal conditions.

These differences in the developmental processes are responsible for their differences in physiological functions and social roles [7] and specific adaptations to environmental conditions. For instance, drone embryos and larvae have more precise temperature regulation compared with workers, and drone pupae are more tolerant to temperature variability [8]. Therefore, understanding the transcriptomic differences between worker and drone embryos could provide a better understanding of the molecular mechanisms underlying their developmental disparities. This could provide valuable insights for optimizing breeding programs by selecting desirable traits and developing targeted conservation strategies to enhance colony resilience and adaptability.

Significant advances have been made in understanding the molecular mechanisms underlying honeybee embryo development, including the visualization of the key stages of embryogenesis and the development of techniques for genetic and molecular manipulation [9,10,11]. The differentially expressed genes between workers and drones (*Apis mellifera* Linnaeus, 1758) during embryo development are sex-specific [4]. The doublesex gene plays a key role in sex determination and reproductive development and influences physiological and behavioral differences between workers and drones [12]. Studies on transcription factors and regulatory networks have identified gene expression patterns influencing sex determination and cell fate and have provided new insights into the relationship between honeybee ontogeny and social behavior [13]. However, the understanding of the molecular characteristics of workers and drones during development is still limited.

Alternative splicing generates different messenger RNA (mRNA) splice isoforms from precursor mRNAs. Alternative splicing is ubiquitous in eukaryotic organisms and significantly enhances the diversity of transcriptomes and proteomes by generating multiple mRNA isoforms from a single gene. Thus, it is a critical post-transcriptional regulatory mechanism in eukaryotes and a key contributor to protein diversity, playing an essential role in phenotypic variation in honeybees [14,15]. Differences in sex-related phenotypes are regulated by selective splicing. For example, male and female somatic cells in *Drosophila* are determined by the selective splicing of the *dsx* gene [16]. In honeybees, genetic mapping and chain analysis have revealed that sex is determined by the heterozygosity of a single gene, the complementary sex determiner (*csd*) gene. Heterozygous individuals develop into females, while hemizygous or homozygous individuals develop into males or diploid males, respectively [17]. Sex-specific splicing was confirmed in the adult stage of honeybees by examining the expression of *dsxF1* and *dsxF2* in female samples and *dsxM* in male samples [18], and selective splicing differences are common between the sexes [19,20].

The previous assembly annotations of the *A. cerana* genome were based on short-read-length sequencing (RNA-Seq), which may have led to underannotated transcripts [3]. Some genes, particularly the different splicing isoforms of sex-determination-related genes, may have been overlooked, and long-read-length sequencing could be used to obtain full-length complementary DNA (cDNA sequences), which could provide more accurate and diverse transcripts [21,22,23]. PacBio Iso-Seq and Illumina RNA sequencing are used for biological transcriptome research [11,24]. Illumina RNA sequencing technology has been used to study the transcriptomic elements and dynamics of embryonic development in the honeybee genome [3]. Iso-Seq produces longer reads, offering more complete information and identifying complex splicing events and structural variations [25]. Iso-Seq technology does not require transcript assembly, which reduces the likelihood of errors, which can occur during the assembly of RNA-Seq short-read fragments. Many extensive transcriptome studies have combined Iso-Seq and RNA-Seq, such as in actinopterygian fish (e.g., *Gasterosteus aculeatus* Linnaeus, 1758) [26], mammals (e.g., goats) [27], and invertebrates (e.g., *Sphingonotus tsinlingensis* Zheng, Tu & Liang, 1963) [28]. The integration of these two sequencing technologies could provide more comprehensive information on the honeybee transcriptome and reveal differences between drones and workers in the honeybee society.

To understand the growth and development patterns of drone and worker embryos and the expression characteristics during five developmental time points (12, 24, 36, 48, and 60 h), long-read (PacBio Iso-Seq) and short-read sequencing (Illumina RNA-Seq) were integrated to investigate the genes and gene expression patterns. Quantitative analysis was conducted on the embryo transcriptome using Illumina RNA-Seq to verify the alternative splicing events of sex- and phenotypic-related genes and changes in gene expression at the five developmental time points. The findings will enhance the understanding of *A. cerana* sexual differentiation during embryonic development and provide basic data on the transcriptome landscape of honeybee embryogenesis. Additionally, they will provide valuable insights into broader biological processes regulated by alternative splicing.

## 2. Materials and Methods

### 2.1. Sample Collection

*A. cerana* was collected from the Yunnan Cangyuan Experimental Station of the Institute of Apicultural Research, Chinese Academy of Agricultural Sciences, Cangyuan County, Yunnan Province. Five *A. cerana* colonies were randomly selected, and the mated sister queens were confined to egg-free frames for 6 h before being placed back into the hives. From each colony, 20–40 developing embryos were collected at 12, 24, 36, 48, and 60 h after egg laying. The samples were treated with lysate and stored at −80 °C. A total of 30 samples (each with three biological replicates) were collected for Illumina RNA-Seq and PacBio Iso-Seq.

### 2.2. Total RNA Extraction and PacBio Iso-Seq and Illumina RNA-Seq

The total RNA was extracted from embryo samples stored in PRL lysate using the Flash Pure Total RNA Micro Kit (Cat, R516-PRL, Beijing, China) and following the manufacturer’s protocol. Then, it was qualified and quantified as follows: (1) the RNA purity and concentration were examined using a NanoDrop 2000 (Thermo Fisher Scientific, Waltham, MA, USA) and (2) the RNA integrity and quantity were measured using the Agilent 2100/4200 system (Agilent Technologies, Santa Clara, CA, USA).

An RNA library for RNA-Seq was prepared as follows. The mRNA was purified from the total RNA using Poly A and then fragmented into 300–350 bp fragments. Then, the first-strand cDNA was reverse-transcribed using fragmented mRNA and deoxynucleotide triphosphates (dNTPs; dATP, dTTP, dCTP, and dGTP), and second-strand cDNA synthesis was performed. The remaining double-strand cDNA overhangs were converted into blunt ends via exonuclease/polymerase activities. After the adenylation of the 3′ ends of the DNA fragments, the sequencing adaptors were ligated to the cDNA, and the library fragments were purified. The template was enriched by polymerase chain reaction (PCR), and the PCR products were purified to obtain the final library. The sequencing libraries were generated using the VAHTS mRNA-seq v2 Library Prep Kit for Illumina (Vazyme, Nanjing, China) following the manufacturer’s recommendations, and index codes were added to identify the sequences. After library construction, the concentration of the library was measured using the Qubit 2.0 Fluorometer (Life Technologies, Carlsbad, CA, USA). The size distribution of the library was detected by agarose gel electrophoresis. After library preparation and pooling of the different samples, the samples were subjected to Illumina sequencing.

A total amount of 2 µg of RNA was used for sample preparation. PacBio single-molecule real-time (SMRT) bell library preparation was performed using the SMRTbell Express Template Prep Kit 2.0 (PN 101-853-100, Pacific Biosciences, Menlo Park, CA, USA) following the manufacturer’s instructions (Procedure & Checklist-Iso-Seq Express Template Preparation for Sequel and Sequel II Systems). In brief, (i) the first strand of cDNA was synthesized by reverse transcription after adding primer with Oligo dT paired with Poly (A). (ii) Then, template switching was performed by adding the other primer paired with TS oligo. (iii) Subsequently, full-length cDNA was obtained by cDNA amplification with barcoded cDNA PCR primers. (iv) The amplified cDNA was purified with ProNex Beads and quantified by a Qubit 2.0 Fluorometer (Life Technologies). (v) Next, the SMRTbell library was prepared using Kit 2.0. (vi) Then, single-strand overhangs were removed, and DNA damage reparation, end-repair, A-tailing, and adapter ligation were conducted. After library purification using 0.45X AMPure PB beads and 1–10 Kb size selection, the library was prepared for sequencing on the Sequel II system (Pacific Biosciences).

### 2.3. Long-Read RNA Alignment and Isoform Identification

The worker and drone embryo samples produced an average of 1.24 million original sub-read segments (15.3 GB in total). Circular consensus sequences (CCSs) were created using ISOseq3 (v4.0.0; https://github.com/ylipacbio/IsoSeq3, accessed on 3 July 2024), and the barcode and primer sequences were removed by lima (v2.7.1; https://github.com/PacificBiosciences/barcoding, accessed on 3 July 2024) to generate FLNC sequences. These were compared to the reference genome (GCA_029169275-1) using pbmm2 (v1.13.1), and the transcript was generated and incorporated into isoforms by TAMA (v1.1.2) [29]. Then, SQANTI3 (5.1.1) was used to compare and classify the isoforms, filter false positive transcripts, and generate descriptive charts [30,31]. To maximize transcriptional identification, the total RNA was extracted from the worker and drone embryos of *A. cerana* at five different developmental time points (12, 24, 36, 48, and 60 h). The RNA samples of the worker and drone embryos at each time point were combined to construct two independent libraries for Iso-Seq. Each sample generated an average of 1,242,655 CCSs from the original segments, which were then filtered and clustered into full-length transcripts, generating an average of 703,781 high-quality consensus segments per sample (Appendix A).

### 2.4. Mapping and Quality Control of the Illumina RNA-Seq Data

Initially, the raw data (raw reads) in FASTQ format were processed using fastp (0.23.2). Clean reads were obtained by removing (1) reads containing adapters, (2) reads with more than 3 bases, and (3) reads with more than 20% bases with a Qphred ≤ 5 (Ref). Then, the Q20, Q30, and GC content of the clean data were calculated. Paired-end clean reads were then aligned to the reference genome of *A. cerana* (GCA_029169275.1) using Hisat2 (v2.2.1) [32]. Samtools (v1.6) was used to compare and sort the aligned SAM files into BAM files [33]. FeatureCount (v2.0.1) was used to count the number of reads mapped to each gene [34]. Finally, the Trinity (v2.15.1) software was used to perform differential expression analysis using the DESeq2 algorithm [35].

### 2.5. Gene Ontology and Kyoto Encyclopedia of Genes and Genomes Analysis

Emapper.py (2.1.10) was used to annotate the longest protein sequence in the reference of *A. cerana* genome using eggNOG database annotation, and the annotated file was converted into the org.db format for the gene ontology (GO) and Kyoto Encyclopedia of Genes and Genomes (KEGG) enrichment analysis using the clusterProfiler package in R [36].

### 2.6. Quantitative Real-Time Polymerase Chain Reaction

We employed quantitative real-time PCR (qRT-PCR) technology to quantify the expression levels of alternatively spliced variants of the target gene. To eliminate genomic DNA contamination and ensure RNA purity, a 5 × gDNA Clean Reaction Mix (Hunan Accurate Biotechnology Co., Ltd., Changsha, China, Catalog No. AG11728) was used. This kit effectively removes genomic DNA from RNA samples, preventing DNA contamination in subsequent reverse transcription and real-time PCRs and ensuring the accuracy and reliability of the samples. For the reaction system, 2.0 μL of 5 × gDNA Clean Reaction Mix was used, and the reaction was conducted at 42 °C for 2 min. The RNA template amount was less than 1000 ng, and RNase-Free water (Beijing Solarbio Science & Technology Co., Ltd., Beijing, China) was added to bring the total volume to 10 μL. Subsequently, the extracted RNA was reverse-transcribed into cDNA using a reverse transcription kit (Hunan Accurate Biotechnology Co., Ltd., Catalog No. AG11728). The reverse reaction was performed at 37 °C for 15 min, which was followed by a heat treatment at 85 °C for 5 s.

For the qRT-PCR amplification, a qRT-PCR kit (catalog number: AG11739; Hunan Accurate Biotechnology Co., Ltd.) was used for qRT-PCR. The qRT-PCR system included 2 × SG Green qRT-PCR Mix, forward and reverse primers (each 0.2 μL), and 1 μL of cDNA. The reaction conditions included an initial denaturation at 95 °C for 2 min, followed by 40 cycles of denaturation at 95 °C for 5 s, and an extension at 60 °C for 30 s.

Data analysis was performed using the 2^−ΔΔCT^ method, with internal reference genes (Actin) employed to correct for differences in RNA concentration and reverse transcription efficiency between the samples. Subsequently, *t*-tests were conducted to evaluate the differences in the relative expression levels of the target gene among the different experimental treatment groups, and the relative expression levels for each group were calculated (Appendix A).

## 3. Results

### 3.1. Comparative Analysis of Sex-Specific Splicing and Gene Expression

The selective splicing difference analysis of the sex genes in the honeybee embryos identified specific shearing events in both embryos at five time points (Figure 1a). To improve the accuracy of gene annotation using Iso-Seq, isoforms with expression levels below 15 were filtered out. Of the isoforms expressed in embryos of both sexes (total 59,254), 5744 (7%) were worker-specific isoforms, while 5106 (6%) were drone-specific isoforms (Appendix A). In comparison with the reference annotation, the 24-h stage exhibited the largest disparity in the number of new genes and splice events, especially for workers (Figure 1b). Compared with the reference gene, the stage-specific expression of the *csd* gene was identified at different developmental stages, and a novel isoform was identified (G109027.21). After filtering out the low expression values in the RNA-Seq data, the expression differences were analyzed (Figure 1c,d). The results showed that the isoform *G10927.1* exhibited the highest expression level in the 36-h embryos, while it was significantly suppressed in the drone embryos across all developmental stages. Conversely, the isoform *G10927.39* reached its peak expression in the drone embryos at the 48-h stage (Figure 1c, Appendix A). Notably, the overall expression of the novel isoform *G109027.21* was higher in the drones than in the worker embryos.

### 3.2. Analysis of Differentially Expressed Genes

Through RNA-Seq, 30 libraries generated 2,984,509,350 raw reads, with an average of 99,483,645 reads per sample. After quality control, a total of 2,970,242,921 clean reads were obtained, with an average of 99,008,097 reads per sample. The average quality scores for the clean reads, Q20 and Q30, were 98.87% and 96.38%, respectively, indicating high data quality (Appendix A). The sliding window density analysis showed that more than 94% of the transcripts were uniquely mapped to the reference genome (Appendix A). Among the five time points, the most significant gene expression differences between the sexes were observed at 24 h, with 3391 differentially expressed genes identified (Figure 2a). After filtering out the low-expressed genes and adjusting the *p*-values (padjust > 0.05 and |logFC| > 1), the volcanic maps of the embryonic development were drawn to show the number of up-regulated, down-regulated, and non-significant genes at each stage. The 10 genes with the most significant *p*-values in each of the five time points were ranked, and a large number of new genes or new isoforms of differential expression were identified. In the drone and worker embryos, the 24-h stage exhibited the highest number of differentially expressed genes, reaching 3391, while the fifth stage had the lowest number of significant differentially expressed genes, at 207 (Figure 2b).

Alternative splicing analysis verified the expression levels of the different gene transcripts at the five time points of embryonic development, and the number of alternative splicing genes and events at the five time points of embryonic development significantly overlapped with the distribution of differential gene expression. At the 24-h stage of embryonic development, the number of alternative first exon events reached its peak at 215, accounting for 45% of all splicing events. Additionally, the highest total number of alternative splicing events occurred at this time point (470; involving 352 genes). In contrast, the stage with the fewest alternative splicing events had only 97 events, corresponding to 78 genes (Figure 2c,d).

### 3.3. Analysis of Gene Ontology Functional Enrichment

The results of the GO enrichment analysis (Appendix A) showed that the number of pathways involving significantly differentially expressed genes in the embryos from the first to the fifth time points were 3, 108, 178, 363, and 54, respectively, with the fourth stage having the highest number of pathways. We selected 60 differentially expressed genes for the GO enrichment analysis (Figure 3a). The results indicated that genes related to sexual development (e.g., mating and male courtship behavior) showed significant differences during the third time point of embryonic development in both worker and drone embryos. Specifically, genes related to gonad development, learning and memory, eye development, muscle development, and neurological development also exhibited significant differential expression at the fourth time point.

To better understand the expression patterns and relationships of the genes across the different samples, the top 30 genes with the highest expression levels were extracted and expression cluster analysis was conducted. The results showed that genes associated with reproductive organ development began to increase in expression at 24 h of embryonic development (Figure 3b). Genes related to eye development showed significant changes at 24 h, and then, their level of expression gradually decreased. Genes associated with growth and development exhibited high specificity at 24 h, indicating that this period is a critical window for developmental regulation.

In this analysis, the clustering focused on five key developmental time points, and the selection of the top 30 genes was based on the highest expression levels and biological relevance across these stages. This approach captured dynamic expression changes and identified genes likely to play crucial roles in the embryonic development of both workers and drones.

### 3.4. Analysis of Sex-Specific Alternative Splicing

To further explore the gene differences between the sexes during embryonic development, we studied the retention intron involved in drone courtship behavior (*Y-y*), *dsx* gene, and *cds* gene in both worker and drone embryos at 24 h (Figure 4a–c). In addition, variable shear differences in brain development (*LOC725754*) and eye development (*LOC102654776*) were also identified (Appendix A). Additionally, quantitative PCR (qPCR) experiments were conducted to validate the alternative splicing of these genes. The results showed that the specific spliced forms of the *dsx*, *csd*, and *Y−y* genes were significantly more highly expressed in drones than in workers (Figure 5).

## 4. Discussion

Embryonic development is a crucial stage in the development of insects, with gene expression during the embryonic period playing an important role in ensuring normal development throughout their life cycle [4,37,38,39]. Iso-Seq technology enables the direct reading of full-length transcripts, avoiding the uncertainties associated with transcript assembly in RNA-Seq and significantly improving the accuracy of genome annotation [40]. This technology precisely identifies splicing variants and new splice isoforms and provides richer functional information [40]. Furthermore, the combination of Iso-Seq and RNA-Seq integrates the quantitative advantages of short-read data with the structural integrity of long-read data, allowing for the more accurate analysis of gene expression patterns and diversity. This mixed sequencing strategy has achieved higher-precision annotations in multiple species, including *Ptychobarbus chungtienensis* (Tsao, 1964) [41], *Ovis aries* Linnaeus, 1758 [42], *Gossypium purpurascens* Poir [43], *Procambarus clarkii* (Girard, 1852), and pigs [44].

In this study, we comprehensively analyzed gene expression differences in embryonic development between workers and drones using PacBio Iso-Seq and Illumina RNA-Seq to enhance the understanding of sex differentiation during embryonic development in *A. cerana*. Transcriptome analysis across five developmental time points (12, 24, 36, 48, and 60 h) revealed that the 24-h stage is critical for gene differential expression and alternative splicing events during the embryonic development of workers and drones. In particular, during embryonic development, novel isoforms of the *csd* gene (G10927.21) were identified. Through qPCR validation, the existence of these newly identified isoforms was confirmed (Figure 5, Appendix A). These findings provide a foundation for the functional study of the *A. cerana* genome [45].

Based on Iso-Seq, specific transcripts and novel genes peaked after 24 h of embryonic development, and then, their expression gradually decreased. This trend was consistent with changes in the number of differentially expressed genes and alternative splicing events observed with RNA-Seq. Additionally, the principal component analysis plot indicated that the RNA-Seq data had good reproducibility (Appendix A), further supporting the reliability of the data. These findings reveal the complexity of gene activity after 24 h of embryonic development in *A. cerana*. Iso-Seq transcriptome annotations showed that more than 7% of the variable transcripts were only present in drones or workers (Appendix A). This suggests that, in addition to sex-biased gene expression, sex-specific isoforms may be a common mechanism for achieving sex-specific functions, which was similarly reported in three spine stickleback fish [26]. During this development stage, we identified unknown functional genes that may play a key role in sex differentiation in *A. cerana*.

The differences in the number of isoforms and the overlapping distribution of their corresponding genes across the five developmental time points further explained the role of the different isoforms in gene and functional expression. These isoforms may contribute to the formation of morphological and behavioral differences between adult drones and workers, and at the embryonic stage, genes related to these characteristics exhibited differential expression. The ten most significantly different novel genes at each developmental stage were screened, and 12 new isoforms were identified. Additionally, there were significant trends and quantitative differences in isoform expression between embryos of different sexes. For example, the two novel isoforms, *novelGene_11834* and *novelGene_7210*, exhibited significantly higher expression levels in worker embryos at 24 h compared with drones. Although the specific functions of these new genes remain unclear, their significant expression differences between worker and drone embryos suggest that they may play important roles in sex-specific development. The *novelGene_11834* gene was significantly more expressed in worker embryos at 24 h compared with drones, and *novelGene_7210* also showed significantly higher expression in workers than in drones. These findings suggest that these genes may be involved in the formation of the unique physiological and morphological characteristics of workers (Appendix A) [18]. A further analysis of the functional annotations and associated pathways of these novel genes may help with a better understanding of their roles in sex differentiation.

Differences in alternative splicing have been reported to significantly impact sex differentiation and development [46,47]. Moreover, numerous studies have shown that sex-specific isoforms play key roles in honeybees [18], *Drosophila* [16], *Artemia franciscana* Kellogg, 1906 [47], birds [20], and zebrafish [48]. In *Drosophila*, sex-specific differences in transcript diversity are also present during embryonic development [49]. Although the functions of these isoforms have not yet been clearly identified, their significant differences in sex-specific expression provide important clues for future functional studies. In this study, through detailed expression analysis, we identified the alternative splicing forms of key sex-determination genes, such as *csd* and *dsx*, at the 24-h developmental stage. This indicates that sex-specific alternative splicing events occur as early as the embryonic stage and may play a crucial role in initiating sex differentiation. The presence of distinct splice isoforms of key sex-determination genes further suggests that alternative splicing serves as a central regulatory mechanism driving sex differentiation in *A. cerana*. Based on these findings, further investigation into the functional roles of these sex-specific isoforms could provide new insights into the molecular mechanisms underlying sex determination and differentiation in *A. cerana*. Furthermore, this regulation may affect genes related to eye structure, potentially resulting in differences in compound eye morphology between drones and workers (Appendix A). Understanding how these isoforms influence developmental pathways could enhance the understanding of honeybee development and provide insights into their developmental biology and genetics.

The significance of the yellow gene family in various biological processes has been well documented across multiple species. For example, mutations in the yellow gene family in fruit flies can lead to changes in pigment deposition in the wings, abdomen, and thorax, and can affect behavioral patterns through interactions with the sex determination pathway. Similarly, when the *Yellow-y* gene is knocked out in silkworm larvae, the originally black larvae exhibit yellow or lighter body colors [50,51,52,53]. Additionally, studies have found that the yellow gene family plays a critical role in body color formation in the fall armyworm (*Spodoptera frugiperda* J.E. Smith, 1797) and the black soldier fly (*Hermetia illucens* (Linnaeus, 1758)) [53,54]. Notably, the yellow gene family affects eggshell formation in the red flour beetle (*Tribolium castaneum* (Herbst, 1797)), and these genes are essential during eggshell formation and early embryonic development [52].

In studies on Asian honeybees (*A. cerana*), the yellow gene family was associated with the differentiation of workers and queens [49]. These genes are expressed in the brains, venom glands, and other tissues of larvae, pupae, and adult honeybees [49], indicating that they play important roles in the development and behavior of honeybees. The *Yellow-y* gene in honeybees is believed to be closely related to sex differentiation, which is consistent with research findings in other species. In this study, we confirmed the expression of different isoforms of the yellow gene family in 24-h-old embryos of both sexes (Figure 5, Appendix A). This suggests that the *Yellow-y* gene may be involved in sex differentiation, wing development, and pigmentation differences during embryonic development.

However, the study has certain limitations, such as the inability to fully explore the functional mechanisms of the *Yellow-y* gene in sex differentiation and development due to the lack of experimental validation. Future research could focus on deeper molecular analyses and the application of gene knockdown or overexpression techniques to further elucidate the role of the yellow gene family in honeybee development and behavior.

## 5. Conclusions

In conclusion, to explore the molecular differences between the transcriptomes of worker and drone embryos in *A. cerana*, PacBio Iso-Seq and Illumina RNA-Seq were conducted, and the 24-h development time point was found to be the most significant stage of gene differential expression and alternative splicing events during the development of worker and drone embryos.

This study provides new data for genome annotation of *A. cerana* and reveals the important role of sex-specific transcripts in the sex differentiation of *A. cerana* at the embryo stage. These findings lay the foundation for future studies of honeybee transcriptomics and gene regulation and provide insights into the molecular mechanisms of honeybee sex differentiation.

## Figures and Tables

**Figure 1 genes-16-00187-f001:**
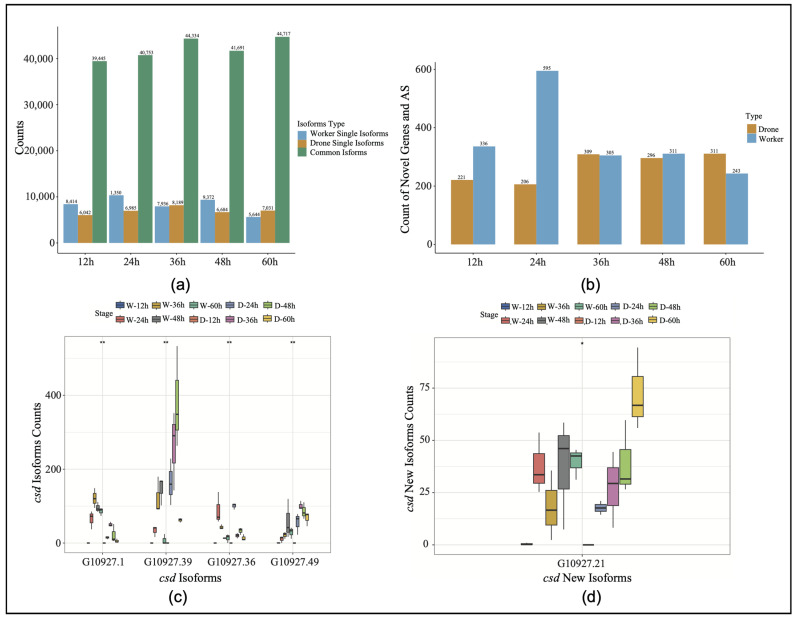
(**a**) Expression levels of specific isoforms and novel genes across five time points. W, worker; D, drone. (**b**) Discovery of novel genes and new splicing events in drones and workers. AS, alternative splicing. (**c**) The expression levels of the different isoforms of the *csd* gene (* *p* < 0.05, ** *p* < 0.01). (**d**) The expression levels of the various novel *csd* isoforms (* *p* < 0.05, ** *p* < 0.01).

**Figure 2 genes-16-00187-f002:**
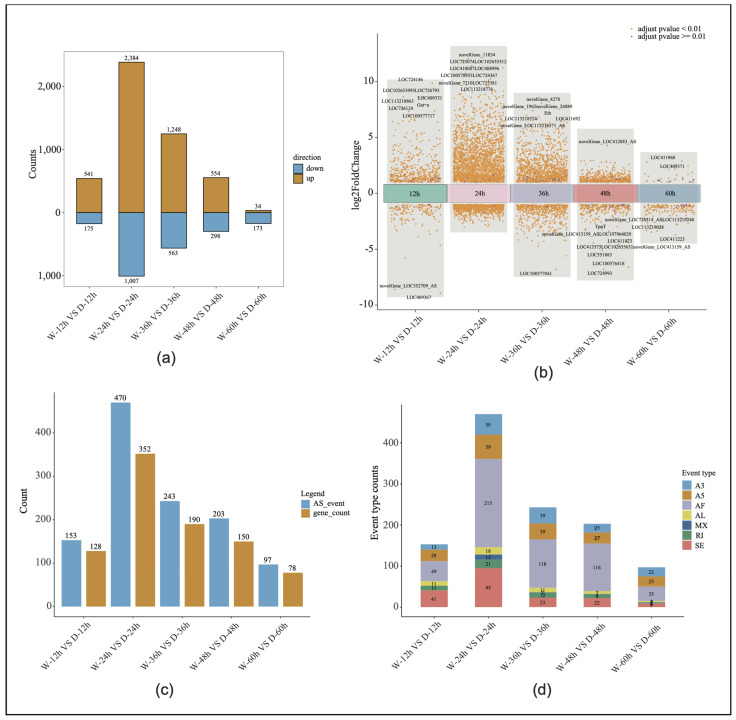
Differential gene expression analysis: (**a**) The number of significantly differentially expressed genes in both drone and worker embryos across five developmental time points. (**b**) Volcano plots depicting the differentially expressed genes in both drone and worker embryos at five developmental time points. W, worker; D, drone. (**c**) Variable splicing events and the corresponding gene count at five time points during worker and drone embryonic development. AS, alternative splicing. (**d**) Graph showing the number of seven types of alternative splicing events across five time points of worker and drone embryonic development.

**Figure 3 genes-16-00187-f003:**
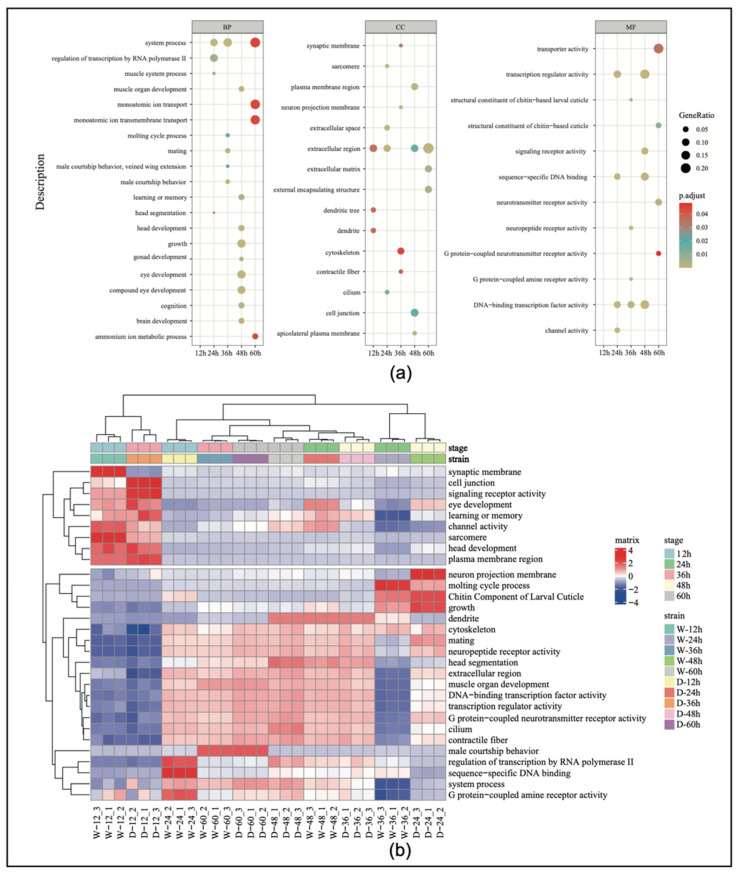
(**a**) Gene ontology (GO) enrichment categorization chart during embryonic development. (**b**) The heatmap of the top 30 genes by expression level in the GO enrichment analysis. BP, biological process; CC, cellular component; MF, molecular function.

**Figure 4 genes-16-00187-f004:**
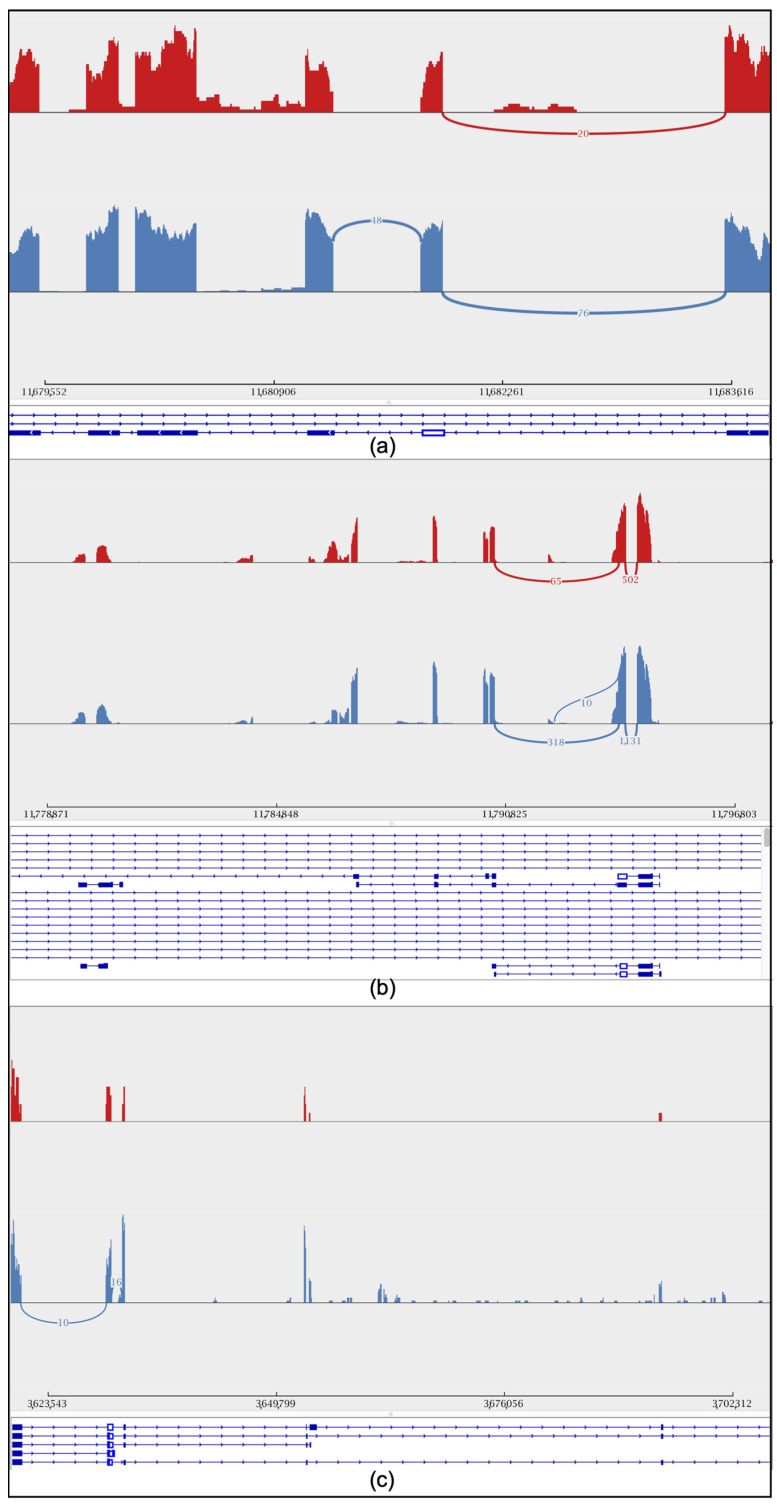
Sashimi plot showing the intron retention level of the sex gene. Peaks in red represent short-read coverage. For each isoform, blocks in blue represent exons, and lines between blocks represent introns. The red color represents workers, while the blue color represents drones. (**a**) Drone courtship behavior gene (*Y-y*). (**b**) *csd* gene. (**c**) *dsx* gene.

**Figure 5 genes-16-00187-f005:**
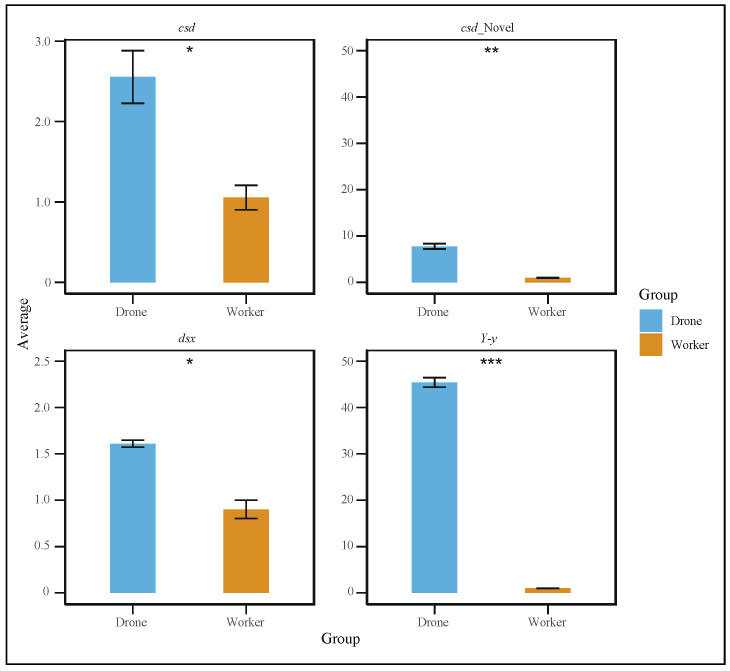
Quantitative polymerase chain reaction (qPCR) results showing that the specific spliced forms of the *dsx*, *csd*, and *Y−y* genes were significantly more highly expressed in drones than in workers (*t*-test, *p* < 0.05). (* *p* < 0.05, ** *p* < 0.01, *** *p* < 0.001).

## Data Availability

The Iso-Seq and short-read RNA-Seq data involved in this study were submitted to the NCBI BioProject database (https://www.ncbi.nlm.nih.gov/bioproject/ accessed on 22 January 2025), with accession numbers PRJNA1186594 and PRJNA1186919. All the custom scripts are provided as supplementary code and can be found on GitHub (https://github.com/Surunlang/Su/tree/main accessed on 22 January 2025).

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
