# Peer review of "Transcriptomic Analysis Reveals Patterns of Expression of Stage-Specific Genes in Early Apis cerana Embryos"

_genes, 2025, doi:10.3390/genes16020187_

Round 1

Reviewer 1 Report (Previous Reviewer 2)

Comments and Suggestions for Authors

This version of the manuscript is much improved over a previous version.  There are still a few (relatively minor) points I would like to see addressed before publication

1.  In the title, I suggest replacing the word "significant" with "patterns of".  I think this title: "Transcriptomic analysis reveals patterns expression of stage-specific genes in early Apis cerana embryos" more correctly summarizes the work done here.  

2. In several places, where dogmatic statements like: "Apis cerana development comprises four stages: embryo, larva, pupa, and adult."  Since this is a somewhat arbitrary way to describe developmental stages, I strongly suggest that this statement be modified as follows: "Apis cerana development be described as comprising four stages: embryo, larva, pupa, and adult."

3. In summarizing the results shown in Figure 4, the statement is made (in lines 316-318): ...The results demonstrated that intron retention and alternative splicing play a key role ...", but no specific result is described to support this statement.

Author Response

Comments 1: In the title, I suggest replacing the word "significant" with "patterns of".  I think this title: "Transcriptomic analysis reveals patterns expression of stage-specific genes in early Apis cerana embryos" more correctly summarizes the work done here.

(Reviewer #1)

Response 1: Thank you for your thoughtful comments. I believe they are very valid.

Modified as follows:

Transcriptomic analysis reveals patterns of expression of stage-specific genes in Apis cerana embryos

Comments 2: In several places, where dogmatic statements like: "Apis cerana development comprises four stages: embryo, larva, pupa, and adult."  Since this is a somewhat arbitrary way to describe developmental stages, I strongly suggest that this statement be modified as follows: "Apis cerana development be described as comprising four stages: embryo, larva, pupa, and adult."

 (Reviewer #1)

Response 2: Thank you for your thoughtful comments. I believe they are very valid.

Modified as follows:

Apis cerana development be described as comprising four stages: embryo, larva, pupa, and adult. (lines 12-13)

The life cycle of honeybees development includes four stages: egg, larva, pupa, and adult [1, 2]. (lines 31-32)

Comments 3: In summarizing the results shown in Figure 4, the statement is made (in lines 316-318): ...The results demonstrated that intron retention and alternative splicing play a key role ...", but no specific result is described to support this statement.

(Reviewer #1)

Response 3: Thank you for your thoughtful comments. I believe they are very valid. Thank you for your suggestion. Since there is no specific result to support this statement, I have removed it. Thank you very much !

Reviewer 2 Report (Previous Reviewer 1)

Comments and Suggestions for Authors

I have already reviewed this manuscript. It has been well revised based on the reviewers' comments.

The only change the author needs to make is to add error bars to Figure 5.

Author Response

Comments 4: I have already reviewed this manuscript. It has been well revised based on the reviewers' comments.The only change the author needs to make is to add error bars to Figure 5. (Reviewer #2)

Response 4: Thank you for your feedback. I have revised the image and added error bars based on your suggestion.

This manuscript is a resubmission of an earlier submission. The following is a list of the peer review reports and author responses from that submission.

Round 1

Reviewer 1 Report

Comments and Suggestions for Authors

The manuscript written by Su et al. provided gene expression changes at different developmental stages through transcriptome analysis of Apis cerana embryos. Long-read and short-read sequencing data provided various information regarding splicing and expression patterns. Most of the parts are understandable, but qPCR confirmation needs to be reconsidered.

Major comments:

The author thinks that it is necessary to consider whether qPCR data is biased by the reference gene. When the expression level of the reference gene is different in the drone and the worker, it is not possible to compare the worker and the drone. Even though the author used the same reference gene, their expression level is distinctly different and not comparable between different species.

You may check the expression level of certain genes via different time points.

That would be the reason why you tried different reference genes.

- **Need to check whether the reference gene is RPL32 or Actin**

In m&m, it is written as RPL32, but in supplemental data, it is written as Actin.

Minor comment: italicized materials are not clearly explained.

Line 60: Apis mellifera   italic

Line 134:  poly T à Poly A

Line 128: theFlash space.

Line 129: Beijing CN -- China

  • Adapter and adopter are used interchangeably, but if they are considered synonyms, it seems necessary to unify them.
  • Honeybee and bee as well
  • base pairs and bp
  • hour and h

Line 140: Library information needed: Which kit do you use? ex) TruSeq3

Line 195: cDNA using the reverse transcription kit needs to be specific

Line 142: Qubit® fluorometer  need to be specific such as country.

Line 154: (Invitrogen), need to be specific such as country

Line 166: Finally, SQANTI3 was used to SQANTI3 version needs to be provided

Line 193:  5×gDNA Clean Reaction Mix   need to be specific

Line 219: , 3031    3.031   48,706

Line 230: Figure 1, a and b should be Figure 1a and 1b

Line 248: 16,764

Figure 4 legend is missing.

Expression Levels of specific isoforms  L no capitalization

W represents worker, D represents drone. Repetition

Line 298:  S3).Among the five time  space

P-values needs to be p-values

Figure 8 legend: pcr à qPCR

Italic

Line 387: Ptychobarbus chungtienensis

Ovis aries

Procambarus clarkii

Line 392 396

Apis crana

Line 476

Spodoptera frugiperda

Hermetia illucens

Line 479

Apis cerana

Line 485 Line489: yellow gene

Comments on the Quality of English Language

italic please

Reviewer 2 Report

Comments and Suggestions for Authors

This manuscript represents a lot of work, but in its current form it is impossible to properly review.

What is the overall goal/focus of this work? The abstract lists too many different objectives including (1) a comparison of two transcriptome analysis methods, (2) identification of differentially expressed genes (DEGs), (3) highly detailed and confusing analysis of isoforms that may or may not correspond to real gene products (4) differential expression of genes as a function of caste.

All of this is way to much for one paper - needs to be refocused on one or a very small number of issues. 

Also, much of the terminology used is classic undefined jargon and is confusing.  This makes many of the data contained in the Figures and Tables too hard to understand

Comments on the Quality of English Language

Overall, the English language usage is not acceptable - you must have a native English speaker help with this.

Author Response

Comments 1:What is the overall goal/focus of this work? The abstract lists too many different objectives including (1) a comparison of two transcriptome analysis methods, (2) identification of differentially expressed genes (DEGs), (3) highly detailed and confusing analysis of isoforms that may or may not correspond to real gene products (4) differential expression of genes as a function of caste.(Reviewer #2)

All of this is way to much for one paper - needs to be refocused on one or a very small number of issues. 

Also, much of the terminology used is classic undefined jargon and is confusing.  This makes many of the data contained in the Figures and Tables too hard to understand

All of this is way to much for one paper - needs to be refocused on one or a very small number of issues. 

Also, much of the terminology used is classic undefined jargon and is confusing.  This makes many of the data contained in the Figures and Tables too hard to understand

Response 1: Thank you for your thoughtful comments. I believe they are very valid. The focus of my work is to update isoforms using third-generation transcriptomics, and to study the differential genes between worker and drone honeybees. At the same time, I have removed sections (previously 3.1, 3.2, and 3.3), which were related to the annotation of third-generation transcriptomics. These sections were irrelevant to the core focus of the study and appeared redundant, thus detracting from the clarity and focus of the paper.

Specifically, this involves the identification of differentially expressed genes (DEGs) and the analysis of alternative isoforms. To highlight this scientific issue, I have streamlined the Abstract, Results, and Conclusion sections to better focus on the differential genes and distinct transcripts between worker and drone honeybees (page 1,lines 12-29 of the manuscript).

Modified as follows:

Abstract:Apis cerana development comprises four stages: embryo, larva, pupa, and adult. There are significant differences between workers and drones in terms of physiological functions and social roles, and the formation of the organ primordia occurs during the embryonic stage. Therefore, the objective of this study is to investigate the differential expression of and alternative splicing of genes in worker and drone embryos may explain this unique developmental pattern. Methods: Long-read sequencing (PacBio Iso-Seq) and short-read sequencing (Illumina RNA-Seq) were used to investigate worker and drone embryo gene expression differences in A. cerana across five developmental points (12, 24, 36, 48, and 60 h). Results: The study identified 59,254 common isoforms, with 5,744 and 5,106 isoforms specific to worker and drone embryos, respectively. Additionally, a new transcript of the csd gene was identified. The number of differentially expressed genes (3,391) and differential splicing events (470 genes) peaked at the 24-hour embryonic stage. Differential splicing events of csd, dsx, and Y-y were observed in the worker and drone embryos. Conclusions: The gene expression results indicated that the 24-hour embryonic point is a critical period for the expression of genes related to developmental and behavioral differences between workers and drones. The findings provide a theoretical basis for future research on the developmental differences between workers and drones. 

To ensure the reliability of our data, we performed quantitative alignment using second-generation RNA-seq data (Illumina), as Iso-seq has low throughput and is not suitable for quantification. After quantifying with the second-generation data, we filtered out lowly expressed transcripts. As a result, we consider the data to be reliable.The revision can be found on page 7, lines 235-237 of the manuscript.

Modified as follows:
To improve the accuracy of gene annotation using Iso-seq, isoforms with expression levels below 15 were filtered out. (lines 209-211)

In terms of figures, we have made the following adjustments: "Isoforms" refers to transcripts, "gene" refers to genes, and "Novel" refers to newly discovered isoforms or new genes. For example, in Figure 2d, "Eventype" represents the seven different types of differential splicing events identified using Suppa software.

This revision should better highlight my research focus and clearly present the data.

Comments 2:Overall, the English language usage is not acceptable - you must have a native English speaker help with this.

Response 2:

Thank you for your valuable feedback regarding the quality of the English language in our manuscript. To address this concern, we have engaged Meiji Translation Company to proofread the entire article. The company employed native English-speaking professionals who have carefully revised the text to improve both formatting and expression. We hope that these revisions enhance the clarity and readability of the manuscript.

We appreciate your understanding and consideration.
